# Aqueous Pretreatment of Lignocellulosic Biomass for Binderless Material Production: Influence of Twin-Screw Extrusion Configuration and Liquid-to-Solid Ratio

**DOI:** 10.3390/molecules29133020

**Published:** 2024-06-26

**Authors:** Julie Cavailles, Guadalupe Vaca-Medina, Jenny Wu-Tiu-Yen, Laurent Labonne, Philippe Evon, Jérôme Peydecastaing, Pierre-Yves Pontalier

**Affiliations:** 1Laboratoire de Chimie Agro-Industrielle (LCA), Université de Toulouse, INRAE, Toulouse INP, 31030 Toulouse, France; julie.cavailles@hotmail.com (J.C.); guadalupe.vacamedina@toulouse-inp.fr (G.V.-M.); laurent.labonne@toulouse-inp.fr (L.L.); philippe.evon@toulouse-inp.fr (P.E.); jerome.peydecastaing@toulouse-inp.fr (J.P.); 2eRcane, Sainte-Clotilde, 97490 La Réunion, France; jenny.wutiuyen@ercane.re

**Keywords:** biomass pretreatment, twin-screw extrusion, sugarcane bagasse, L/S fractionation, binderless material

## Abstract

This study was carried out to investigate the continuous aqueous pretreatment of sugarcane bagasse (SCB) through twin-screw extrusion for a new integrated full valorization, where the solid residue (extrudate) was used for the production of bio-based materials by thermocompression and the filtrate for the production of high-value-added molecules. Two configurations, with and without a filtration module, were tested and the influence of the SCB composition and structure on the properties of the materials were determined. The impact of the liquid-to-solid (L/S) ratio was studied (0.65–6.00) in relation to the material properties and the biomolecule extraction yield in the filtrate (with the filtration configuration). An L/S ratio of at least 1.25 was required to obtain a liquid filtrate, and increasing the L/S ratio to 2 increased the extraction yield to 11.5 g/kg of the inlet SCB. The extrudate obtained without filtration yielded materials with properties equivalent to those obtained with filtration for L/S ratios of at least 1.25. Since the molecule extraction process was limited, a configuration without filtration would make it possible to reduce water consumption in the process while obtaining high material properties. Under the filtration configuration, an L/S ratio of 2 was the best tradeoff between water consumption, extraction yield, and the material properties, which included 1485 kg/m^3^ density, 6.2 GPa flexural modulus, 51.2 MPa flexural strength, and a water absorption (WA) and thickness swelling (TS) of 37% and 44%, respectively, after 24 h of water immersion. The aqueous pretreatment by twin-screw extrusion allowed for the overall valorization of SCB, resulting in materials with significantly improved properties compared to those obtained with raw SCB due to fiber deconstruction.

## 1. Introduction

Lignocellulosic biomass derived from many agricultural and forestry residues, such as wood processing or harvested crop by-products, represents one of the most abundant renewable resources on Earth. Among agricultural residues, sugarcane bagasse (SCB) is the fibrous residue obtained after sugarcane juice extraction and is the most important sugar industry by-product, with 0.3 tons of SCB generated per ton of processed sugarcane.

SCB, which is mainly composed of lignocellulosic (cellulose, hemicellulose, and lignin) fibers, could be used for fiberboard production by thermocompression. This type of application has been tested recently with delipidated cake obtained after sunflower oil extraction from whole sunflower plants [1], *Broussonetia papyrifera* [2], sugarcane leaves [3], and bamboo [4]. In addition, SCB-based materials have been produced using thermocompression without any other treatments [5,6]. SCB has high lignin (~20%) and sugar (~60%) contents, which could contribute to self-bonding, thereby offering an opportunity to produce cohesive materials without adding any synthetic resin [7]. However, it was shown that a high hemicellulose content in lignocellulosic biomass was detrimental to the mechanical properties and dimensional stability of the final material in the presence of water [8]. A correlation even indicated that a reduction in hemicellulose content was linked with a reduction in water absorption in the final material [9]. In a previous study, hemicellulose degradation during binderless SCB thermocompression improved the water resistance of the final material by allowing internal chemical reorganization [6]. Nevertheless, the lignocellulosic materials obtained with the best tradeoff between mechanical properties and water resistance had shortcomings in terms of their behavior in the presence of water, with much higher thickness swelling values (41%) than those reported for materials bound with synthetic resin, i.e., <13% [10].

These results raise the question of the impact of partial hemicellulose removal on the properties of binderless lignocellulosic materials. Molecule extraction from SCB could modify its chemical composition and affect the properties of materials obtained from the remaining solid residue. Partial hemicellulose solubilization during SCB pretreatment could reduce the hydrophilic character of the final material while recovering a liquid filtrate rich in extractable biomolecules. The addition of a pretreatment could also enable modification of the lignocellulosic structure by making the bagasse compounds, e.g., lignin, more accessible, thereby facilitating internal reorganization of the final material during thermocompression.

SCB pretreatment has been widely studied for the purpose of obtaining refined cellulose that could be used to produce bioethanol [11,12]. But SCB is also known to contain free sugars and phenolic compounds, i.e., predominately ferulic acid and p-Coumaric acid [13,14], that are covalently linked to polysaccharides by ester bonds and to lignin by ester or ether bonds [15]. Solubilized molecules from SCB could be used to produce high-value-added molecules such as sugars, furfural, and polyphenols, the latter of which are particularly interesting, as they have great potential as antioxidants in the food and nutrition sectors [16]. Liquid hot water treatment is a very well-known process for molecule extraction from lignocellulosic biomass. Hydrothermal treatment involves the use of pressure to maintain water in a liquid state at elevated temperatures, so it is an attractive approach for biomass fractionation since no chemicals are used [17]. Aqueous extraction of SCB using this process at 180 °C for 30 min has been shown to release phenolic compounds (14 g/kg of bagasse initial dry matter, DMi), sugars (156 g/kg DMi), acetic acid (55 g/kg DMi), and furfural (5 g/kg DMi) resulting from sugar degradation [18]. This extraction process was also found to be efficient for hemicellulose recovery (100–150 g of xylose per kg DMi) [19,20]. However, the main drawbacks of this type of process are the long extraction times and the large amount of water required to extract the molecules, with L/S ratios > 10 [21,22].

The extruder could be considered a reactor that combines mechanical, thermal, and chemical actions in a single step, with L/S extraction performed in continuous mode. Twin-screw extrusion is an intensified process that can be used for pretreatment to reduce the L/S ratio during aqueous extraction but also to modify the SCB lignocellulosic structure [23]. This process has already been used for the extraction of molecules such as hemicelluloses from poplar [24], wheat straw [25], steam-exploded corncob [26] and wheat bran [27], pectin from apple pomace [28], vegetable oils from coriander [29,30] and sunflower [31,32,33], proteins from alfalfa [34], phenolic acids from hemp by-products [35], and flavonoids with biostimulatory properties from sunflower stalks and heads [36]. As is the case for liquid hot water treatment, subcritical conditions can be achieved in twin-screw extruders to favor the hydrolysis of ether or ester bonds, solubilize hemicelluloses, and decompose lignin into phenolic compounds. The combination of conveying and reverse screw elements positioned near a filtration section results in effective compression of the mixture, dynamic plug formation, and effective liquid–solid separation due to efficient pressure buildup [30]. L/S extraction via twin-screw extrusion is therefore an interesting solution for recovering molecules of interest from lignocellulosic biomass in a single step. Twin-screw extrusion can also be used without a filtration module to break down the biomass structure and make the cell wall internal components more accessible. It is also expected that this pretreatment would modify the fiber morphology by increasing its mean aspect ratio [37]. To go further, we propose a new approach where the plant material would be pretreated by twin-screw extrusion for an integrated full valorization, where the solid residue would be used for the production of binderless materials and the filtrate for the production of biomolecules.

The aim of this study was to investigate aqueous pretreatment of SCB by twin-screw extrusion in order to modify its structure and composition, and to gain insight into the effects on the material properties. The solid residue fraction (extrudate) was used to produce bio-based materials by thermocompression, while the liquid fraction (filtrate) was recovered to analyze the extracted biomolecule composition. Twin-screw extrusion configurations with and without filtration were compared to assess their impact on the properties of the materials generated from the extrudate, and to determine whether the molecule extraction process had positive effects regarding the material properties. With filtration, the impact of the L/S ratio (0.65–6.00) was studied in relation to the extraction yield in the filtrate, the chemical composition of the extrudates, and the final material properties.

## 2. Results and Discussion

### 2.1. SCB Fractionation by Twin-Screw Extrusion: General Characterization

The overall SCB fractionation results are presented in Table 1. The experimental L/S ratios were close to the targeted values. In the experiments with a filtration configuration (trials 2 to 8), the L/S ratio of 6.21 corresponded to the maximum value that should not be exceeded to maintain a compact and stable dynamic plug, and thus to achieve effective liquid–solid separation. Conversely, an L/S ratio of 0.65 was the minimum ratio allowing a filtrate production (<0.4 kg/h) with negligible dry matter content. At such a low L/S ratio, most of the water was absorbed by SCB, with simultaneous evaporation of a large proportion of the introduced water. Moreover, it was not possible to recover enough filtrate in that case for chemical analysis. For the two trials with an L/S ratio of 0.65 and 0.7, i.e., with and without filtration, the current feeding the motor (related to the energy the extruder required to work) was higher (87–94 A) than in all the other conditions (79–81 A). This could have been related to the increased friction exerted on the material due to the low water inlet flow rate, which led to a higher viscosity of the mixture inside the extruder, especially in the vicinity of the CF2C screws.

The extrudate moisture content peaked without filtration (0.65 L/S ratio) and with filtration (L/S ratio of 1.25), with a 39–40% moisture content. Then it decreased as the L/S ratio increased, up to 32%. As the water quantity increased, the extruder was fuller and the dynamic plug formed upstream from the reverse screw elements was longer, leading to more efficient pressing and to a drier extrudate at the extruder outlet. A previous study showed that high L/S ratios produced better dewatering in the case of alfalfa fibers, with a drier solid residue obtained [38].

Otherwise, the quantity of filtrate recovered gradually increased with the L/S ratio due to the increased water quantity used. However, the percentage of dry matter in the filtrate logically decreased for L/S ratios > 1.25 due to a dilution effect. The dry matter flow rate in the filtrate increased with the L/S ratio up to a value of 3.11, where a maximum of 1.6 kg dry matter/h was obtained. Above this L/S ratio, the dry matter flow rate in the filtrate was stabilized at an average of 1.3 kg/h. It has already been shown—on wood by-products—that extractable yields increase with the L/S ratio due to biomass structural degradation and the increased pressing that occurs in such conditions [39]. Higher extractable release occurred when the fibrous plant cell network was disrupted, with better extractable diffusion occurring in more dilute solvent.

The overall balance, including extrudate and filtrate outputs, showed that the evaporated water quantity (in kg/h) increased slightly with the L/S ratio. Conversely, the proportion of evaporated water, expressed in relation to the total inlet water flow rate, decreased with an increasing L/S ratio, reaching a minimum of 9% for the 5.19 L/S ratio. The increase in water quantity at the extruder inlet contributed to reducing the viscosity of the mixture and therefore the mechanical stress exerted, resulting in less self-heating. It was also more difficult to heat up a larger quantity of water: The temperature in the extruder, especially in the eighth and last module along the barrel, decreased with the L/S ratio, and the proportion of evaporated water was lower at the same time, except under the conditions tested in trial 8 (6.21 L/S ratio). A similar trend was noted with rice straw, with a low L/S ratio resulting in a higher proportion of evaporated water [40].

### 2.2. Filtrate Characterization

#### 2.2.1. Distribution of Dry Matter Recovered from the Filtrate

Centrifugation of the filtrates was necessary to separate the small solid particles (pellet) from the soluble molecules (supernatant). Masses of soluble molecules in the supernatants in proportion to the initial dry matter were therefore logically lower than the dry masses in the filtrates. The results showed that the dry matter recovered in the filtrate increased with the L/S ratio, reaching a plateau at an L/S ratio of 3.11 with 78 g of dry matter recovered per kg of DMi (Table 2). Most of this dry matter (about 80%) was present as fine solid particles in the filtrate pellet. The proportion of dry matter in the supernatant was quite independent of the L/S ratio, with a maximum of 25%. These results indicated that SCB deconstruction required a minimum amount of water, which favored the transmission of mechanical energy to the residue fibers as well as better thermal conductivity and heat transfer in the mixture, thereby amplifying the thermomechanical phenomena. When too much water was used, the mechanical energy was not properly transmitted, as the residue became too fluid. However, under these conditions, although the liquid-to-solid separation was less efficient, soluble molecule extraction was stable, as the diffusion increased while the mechanical action decreased. These results demonstrated that SCB fractionation by twin-screw extrusion was able to solubilize only a small part of the molecules, but it had a substantial mechanical impact on SCB.

#### 2.2.2. Extraction Yield in the Supernatant

Concentrations and extraction yields (expressed in proportion to the initial dry matter) of the supernatants are presented in Table 3. The extractable concentrations gradually decreased with the L/S ratio due to a dilution effect. Thus, the supernatant obtained with an L/S ratio of 1.25 was logically the most concentrated one. Total extraction yields gradually increased from 3.7 to 14.5 g/kg DMi. These values are low compared to the extraction yields obtained from SCB by the hydrothermal process due to the shorter residence time and lower temperature used [18]. Regarding the extraction yields obtained by twin-screw extrusion on lignocellulosic biomass in other publications, their extraction yields were higher and could be explained by the fact that the treated biomass, such as corn cobs [26] and wheat straw with bran [25], were characterized by the absence or low presence of lignin, which served as protection for the plant and limited the extraction of compounds. Additionally, the use of alkaline agents such as sodium hydroxide, known to facilitate the solubilization of hemicelluloses and lignin, also contributed to the higher extraction yields [27,35]. However, a plateau was reached above the 5.19 L/S ratio. In all cases, the solubilized molecule distributions in the supernatant were quite similar, with compounds derived from lignin (~25%), xylose (~25%), glucose (~20%), arabinose (~8%), and polyphenols (4–10%). These results confirmed that changing the L/S ratio altered the extraction mechanisms but did not change the type of molecules extracted. The increase in the ash recovery yield with the L/S ratio could reflect an increase in the solubilization efficiency. This result could be explained by better water impregnation in SCB as the L/S ratio increased. However, at high L/S ratios, the solubilized molecules may not have been properly recovered due to the low liquid–solid separation efficiency during extrusion. Furthermore, Morales-Huerta et al. showed that the amount of extracted hemicelluloses from blue agave bagasse increased with longer residence time [41], and it has also been demonstrated that an increase in the liquid flow rate reduced the residence time [24], resulting in a decrease in the extraction yield.

#### 2.2.3. Chemical Characterization of the Filtrate Pellets

Table 4 presents the chemical composition of raw SCB and the fine particles present in the filtrate pellets generated after centrifugation. The chemical compositions of the filtrate pellets differed from that of raw SCB. Filtrate pellets exhibited much lower cellulose, hemicellulose, and acetyl contents compared to raw SCB, whereas the ash and extractable contents were much higher. The increase in extractable contents in the filtrate pellets was the sign that some solubilized molecules were not well separated from solid particles during the centrifugation step, i.e., they remained trapped between them. Corbett et al. found that the filtrate pellets mainly consisted of fine solid particles containing more lignin and extractables, while the cellulose and hemicellulose contents were lower compared to the contents found in the fibers in the case of hardwood pulp separated into several particle size fractions [42]. As the L/S ratio increased, cellulose, hemicellulose, and acetyl contents tended to decrease, while the ash content increased. These changes confirmed that at higher L/S ratios there was greater molecule solubilization and less breakdown of the plant. However, for an L/S ratio of 6.21, the hemicellulose and cellulose contents were higher when the L/S ratio limit to maintain the dynamic plug and ensure correct liquid-solid separation had been reached, leading to poorer solubilization of these molecules. At a low L/S ratio, the extruder acted more as a grinder and could produce fine particles through its mechanical action, whereas at high L/S ratios the extruder acted as a thermomechanical contactor and favored solvent impregnation of the material.

### 2.3. Extrudate Characterization

#### 2.3.1. Physical Properties of the Extrudates

The analysis of the particle size distribution of the extrudates is presented in Figure 1, while their SEM images are shown in Figure 2. Extrudates were composed of lignocellulosic fibers of different sizes but also of ball-shaped aggregates, which were sometimes > 4 mm, thus explaining the increase in the proportion of >1 mm particles in the extrudates compared to the raw SCB before pretreatment (Figure 1). In addition, the SEM images (Figure 2a) clearly show the breakdown of SCB following pretreatment by twin-screw extrusion, with significant fiber defibration and individualization. Pretreatment also caused aggregate formation by fiber entanglement.

At similar L/S ratios (trials 1 and 2), the extrudate obtained without filtration exhibited a lower proportion of particles > 1 mm (27%) compared to the extrudate obtained with filtration (52%) and higher proportions of fibers between 0.2 and 0.8 mm and fine particles (<0.2 mm). Without the filtration module, the extrudate moisture content remained higher (40%) since there was no liquid–solid separation. This limited the aggregation, as observed by comparison of the SEM images in Figure 2b–e, which clearly show the formation of smaller aggregates and greater fiber individualization.

Under the filtration configuration, at an L/S ratio of 0.65, the extrudate also contained a higher proportion of particles > 1 mm (52% compared to 20–35% for the other extrudates obtained at higher L/S ratios) and a lower proportion of fibers between 0.2 and 0.8 mm and fine particles (<0.2 mm). This trial led to the production of an extrudate with a greater tendency to form large aggregates. As explained previously, this extrudate had a lower moisture content (only 24%), which could be attributed to the much higher water evaporation (46%) during aqueous pretreatment, which was conducive to aggregation.

With filtration, for L/S ratios > 1.25, the particle size distribution was directly impacted by the L/S ratio. The proportion of 1–2 mm and 2–4 mm aggregates increased with the L/S ratio, while the proportions of 0.2 to 0.8 mm fibers and fine particles decreased. Comparison of the extrudate SEM images showed that the extrudate obtained with an L/S ratio of 2.05 (Figure 2f,g) was more defibrated and had more individual fibers than the extrudate obtained with an L/S ratio of 6.21 (Figure 2h,i). The aggregates obtained with an L/S ratio of 6.21 consisted of intermingled fibers with few individual fibers, whereas it was possible to distinguish the intermingling of individual fibers on the aggregates obtained with an L/S ratio of 2.05. This greater aggregation could be linked to the decrease in the moisture content of the extrudate (39 to 32%) observed with the increased L/S ratio. It was previously mentioned that this was due to greater extruder filling, which favored the formation of a dynamic plug, leading to more efficient pressing and liquid–solid separation. Another recent study by Theng et al. also reported an increase in the proportion of short fibers in a rice straw extrudate when the L/S ratio used in the twin-screw extruder was reduced [40].

The bulk and tapped raw SCB and extrudate densities are shown in Figure 3. The extrudates obtained had low densities, ranging from 48 to 57 kg/m^3^ bulk density and from 79 to 96 kg/m^3^ tapped density, except for the extrudate obtained with filtration at an L/S ratio of 0.65, which had the highest bulk and tapped densities (96 kg/m^3^ and 147 kg/m^3^, respectively). The majority of the extrudates therefore had densities lower than those of the raw SCB (59 kg/m^3^ bulk density and 103 kg/m^3^ tapped density). These results suggest that aqueous pretreatment through twin-screw extrusion under these conditions might have led to fiber deconstruction, resulting in the swelling and expansion of the SCB fiber structure, thus producing “fluffy” extrudates. In contrast, the high densities observed in the trial with filtration at an L/S ratio of 0.65 could be attributed to the increased pretreatment severity, with a higher proportion of evaporated water and lower extrudate moisture content. As seen previously, this specific extrusion condition tended to favor the formation of large and dense aggregates, contributing to the increase in extrudate density. Similar observations have been reported for coriander processing by twin-screw extrusion, with an increase in the extrudate density at a lower L/S ratio [43].

#### 2.3.2. Chemical Composition of the Extrudates

Table 5 shows the chemical composition of raw SCB and extrudates. The extrudate chemical compositions were very similar, with around 40% cellulose, 24% lignin, and 23% hemicelluloses, which indicated that the configuration mode and the L/S ratio had mainly altered the plant fiber reorganization, as shown by the SEM images, and the particle size distribution. The extrudate obtained with and without filtration at an L/S ratio of 0.65 and 0.70 (trials 1 and 2) exhibited slightly lower cellulose and hemicellulose contents and higher ash and water-soluble extractable contents. These differences may be attributed to polysaccharide degradation during the aqueous pretreatment because of its severity. Shear was higher in this condition due to the low amount of inlet water, which induced higher self-heating compared to the results obtained with other L/S ratios.

### 2.4. Production of Binderless Materials by Thermocompression

The visual appearances of the materials made from the raw SCB and extrudates are shown in Figure 4. The material made from raw SCB exhibited a heterogeneous structure with visible fibers and particles of varying sizes on the surface, and the texture was rough and irregular due to the untreated state of the SCB. The materials obtained from the extrudates were more homogeneous and denser than those made from raw SCB, and their textures were smoother. This indicated that pretreatment influenced the SCB structural properties, with partial fiber deconstruction and aggregation of finer particles. However, these materials had edges that were lighter with beige coloring, which could be explained by a difference in the pressure applied within the material during thermocompression due to a non-uniform distribution of the extrudate induced by manual filling of the mold. Modification of the configuration or L/S ratio had no clear impact on the appearance of the materials.

#### 2.4.1. Density and Mechanical Properties of the Materials

Density of the materials obtained from raw SCB and extrudates and their mechanical properties are presented in Table 6. All the materials had a high density (at least 1432 kg/m^3^), which was much higher than most of the materials obtained from lignocellulosic fibers by thermocompression (700–1350 kg/m^3^) [44], so they could thus be classified as hardboards. The main reasons for the density results obtained are the high pressure and temperature applied during thermocompression (102 MPa at 200 °C) compared to the pressures reported in other studies, which were <30 MPa [7,45]. This could also be explained by the high proportion of fine particles in the raw SCB and extrudates (15–35%) that could fill the voids within the materials, thereby increasing the material density [6]. The material obtained with raw SCB had a lower density (1432 kg/m^3^) than those obtained with extrudates (1475–1503 kg/m^3^). This result could have been due to fiber deconstruction during pretreatment, which resulted in readily deformed and compressed fibers and aggregates, and a higher proportion of fine particles, which acted as fillers. The material densities tended to slightly decrease as the L/S ratio increased. The reduction in the density of the materials could be linked to a reduction in the proportion of fine particles in the extrudate when the L/S ratio increased, except for the configuration with filtration and a 0.65 L/S ratio. In this case, the material density was high despite a lower proportion of fine particles since the extrudate initially had a very high density compared to the other extrudates.

Regarding the bending properties, the material obtained with raw SCB exhibited a lower flexural strength (33.2 MPa) and modulus (4.9 GPa) compared to materials obtained with the extrudates. Their flexural strength ranged from 38.2 to 56.0 MPa, whereas their flexural modulus ranged from 6.1 to 7.1 GPa, leading to a 15–69% improvement in flexural strength and a 25–45% improvement in flexural modulus after SCB aqueous pretreatment by twin-screw extrusion due to fiber deconstruction. These mechanical properties were considered high compared to other thermocompressed materials made from various lignocellulosic residues (10–30 MPa flexural strength) [46]. The reported mechanical properties of binderless materials obtained from SCB are lower, with values of 16.6 MPa for the flexural strength and 3.5 GPa for the flexural modulus [5]. These results could be explained by the low density of the materials (800 kg/m^3^) due to the applied pressure. As previously described, the use of high pressure improved the cohesion and mechanical strength of the material through densification. For SCB thermocompression at a comparable pressure, the mechanical properties reported in our previous work reached a maximum of 43.6 MPa for the flexural strength and 6.9 GPa for the flexural modulus [6]. These values are comparable for the flexural modulus, but pretreatment by twin-screw extrusion provided a clear improvement, with a significant increase in flexural strength due to fiber deconstruction and better internal reorganization of the components during thermocompression.

The properties were impacted by the L/S ratio: The flexural modulus obtained with SCB extrudates produced at an L/S ratio above 5.19 was slightly higher (56.0 MPa), whereas the flexural strength of the material made with the extrudate obtained with filtration at an L/S ratio of 0.65 was much lower (38.2 MPa) than those of the other materials (>50 MPa). The lower flexural strength noted in this case could be attributed to the higher extrudate density, the lower proportion of fine particles, and greater particle aggregation phenomenon observed in the particle size distributions and SEM images. These characteristics could have contributed to a reduction in the accessible surface area of the extrudate, implying that there were fewer areas of contact between the fibers and particles, with less self-adhesion during thermocompression and therefore poorer mechanical properties [47]. The configuration without filtration generated an extrudate that yielded a material with bending properties in the same order of magnitude as those generated from extrudates produced with filtration and L/S ratios of at least 1.25. This indicated that the use of the filtration module, with partial removal of hemicelluloses and lignin, did not markedly change the properties of the final material.

#### 2.4.2. Water Resistance of the Materials

The water absorption (WA) and thickness swelling (TS) values of the materials obtained with raw SCB and extrudates after 24 h of immersion in water are presented in Figure 5. The material obtained with raw SCB had higher WA (90%) and TS (95%) values compared to those obtained with extrudates. Their WA ranged from 37 to 54%, with TS ranging from 41 to 57%. Pretreatment reduced the WA and TS of the materials by at least two-fold due to fiber deconstruction and the higher material density. It was reported that binderless materials obtained by thermocompression of SCB at a similar temperature (200 °C) had a higher TS value of 160% [5]. As described for the mechanical properties, this result could be explained by the low pressure applied. The material had a lower density, which facilitated the penetration of water into the material. For SCB thermocompression at a comparable pressure, the TS value reported in our previous work reached a minimum value of 9% [6] due to a higher thermocompression time (30 min) and the particle size of the SCB used, which had been finely ground using a 2 mm sieve. For a thermocompression time of 10 min, the TS value was 47%, which was comparable to the values obtained with the twin-screw extrusion pretreatment.

As for the mechanical properties, the use of a configuration with or without filtration for extrudate production led to the obtention of materials with similar water resistance properties. The L/S ratio only had a minor impact on WA and TS, given that the values were in the same range. The best tradeoff would be to operate without filtration to limit water consumption in the twin-screw extrusion process, or with filtration at a L/S ratio of 2.05 in order to improve water resistance while minimizing the inlet water volume.

#### 2.4.3. Potential Applications

Based on the measured properties, these materials could be considered high-density fiberboards since their density was >800 kg/m^3^ [48]. They could therefore be used for furniture making or building applications. According to standard ISO 16895:2016 [49], which outlines the requirements for fiberboards made from wood fibers, these materials met the criteria for use as hardboards in a dry environment. However, the materials did not meet the criteria for use in wet or outdoor environments. All of the materials had mechanical properties that exceeded the requirements, but for this type of use, the TS value should be at least <20%, which was not the case for the materials obtained under the study conditions. The material closest to this criterion was that made with the extrudates obtained with a filtration configuration and an L/S ratio of 2.05, which had a WA of 37% and a TS of 44%. Under these conditions, pretreatment resulted in a notable improvement in material properties, with a 27% increase in flexural modulus, a 54% increase in flexural strength, a 59% reduction in WA, and a 56% reduction in TS. For outdoor environments, the material dimensional stability should be improved. Several strategies could be considered, including the application of surface coatings (varnish, resin, or drying oil, such as linseed oil) or thermal post-treatments. In a recent study on fiberboards made from sunflower co-products [1], it was shown that a linseed oil coating could reduce TS by 27%, while thermal post-treatments demonstrated the greatest effectiveness, with an 80% reduction in TS. The benefits of thermal post-treatments have also been demonstrated on coriander fiberboards [43].

## 3. Materials and Methods

### 3.1. Raw Material Preparation

Air-dried SCB was provided by eRcane (Réunion, France). SCB was ground using a 16 mm grid with an Electra F6 N V hammer mill (Paris, France) to homogenize the particle size of the raw material before pretreatment. Ground SCB before pretreatment was referred to as raw SCB. The SCB was moistened in a drum mixer by water spray to achieve a water content (*w*/*w*) so as to promote fine particle aggregation with long fibers, thereby avoiding segregation during extruder feeding. For chemical composition characterization, part of the raw SCB was ground, using a 2 mm mesh grid, with an IKA-Werke MF 10 basic Microfine grinder drive (Staufen im Breisgau, Germany).

### 3.2. Twin-Screw Extrusion Pretreatment

SCB was subjected to aqueous pretreatment using a Clextral Evolum HT 53 co-penetrating and co-rotating twin-screw extruder (Firminy, France), with a 53 mm screw diameter (D), to produce the extrudates and filtrates. The extruder had eight modular barrels, each 4 D in length, except for module 1, which had a length of 8 D (i.e., 36 D for the total barrel length). Modules 2 to 8 were heated by electric resistance and cooled by water circulation. The SCB was fed at a rate of 20 kg/h DM into the extruder inlet port using a Coperion K-Tron SWB-300-N constant-weight feeder (Niederlenz, Switzerland) in module 1. Water was injected using a Clextral DKM Super K Camp 112/12 piston pump (Firminy, France) at the end of module 2. The screw profile used in this study is provided in Figure 6. Two extrusion configurations, with and without filtration, were tested. In the filtration configuration, a filter section consisting of six hemispherical grids with a 1 mm diameter mesh was positioned on module 7 to enable filtrate collection.

After liquid injection, a series of BL22-1.0-90° 2-lobe kneading blocks (1 D total length) and two series of BL22-0.5-90° 2-lobe kneading blocks (1 D total length) were placed at the end of modules 3 and 5, respectively, to ensure intimate water diffusion in the SCB. The CF2C-0.5-0.5 reversed double-thread screws (1 D total length) were positioned at the beginning of module 8 to achieve an intense shearing of the mixture for deconstruction and in the presence of the filtration module to form a dynamic plug to ensure pressing of the mixture to recover the filtrate. The screw rotation speed (250 rpm), SCB feed rate, inlet water flow rates, and barrel temperature were monitored with a control panel. The set of barrel temperature values were 25, 25, 75, 100, 100, 100, 100, and 100 °C from modules 1 to 8, respectively. The experimental variables of this study included the extruder configuration (with and without filtration) and the L/S ratio used (0.65 to 6.00). The operating conditions are presented in Table 1.

Twin-screw extrusion was run for 5 min before each sample collection to stabilize the operating conditions of the process. Once these conditions were steady, the extrudate and filtrate were immediately collected over a 4 min period to avoid any variability in the outlet flow rates. The sample collection time was determined with a stopwatch. For each tested condition, sample collection was carried out in duplicate, and the extrudate and filtrate were then weighed. The extrudate moisture content was also measured immediately after its collection with a Sartorius MA35 infrared weighing balance (Göttingen, Germany). Extrudates were then dried at 50 °C for at least 12 h and then conditioned in an environmental chamber at 25 °C and 50% relative humidity (RH) for 3 weeks until their moisture content was stabilized, prior to their characterization and transformation into materials by uniaxial thermocompression. Otherwise, the filtrates were centrifuged at 10,000× *g* at 20 °C for 10 min to separate the remaining solid (pellet) from the extract containing the solubilized molecules (supernatant). The pellet containing fine solid particles was dried at 105 °C and then conditioned in an environmental chamber at 25 °C and 50% RH for 3 weeks prior to chemical characterization. The supernatant was stored at 4 °C prior to chemical analysis.

### 3.3. Analytical Methods

#### 3.3.1. Chemical Composition of the Solid Samples

The chemical compositions of the solid lignocellulosic samples (raw SCB, extrudates, and pellets) were determined in triplicate. The contents are expressed as a percentage of dry matter (DM). The DM content was evaluated by drying the sample at 105 °C until constant weight, and the ash content was measured after mineralization at 550 °C for 12 h [50].

Chemical characterization was carried out using a procedure based on the laboratory analytical procedure of the National Renewable Energy Laboratory (NREL) [51], as also described previously [6]. Briefly, water and ethanol extractable contents were determined after water and then ethanol (96%) extraction using 1 g of sample and 100 mL of boiled solvent for 1 h in a Foss Fibertech FT 122 extraction system (Hillerød, Denmark). Cellulose, hemicellulose, and lignin contents were determined after a two-step hydrolysis process using 72% sulfuric acid from VWR (Radnor, PA, USA) at 30 °C for 1 h and then a 4% sulfuric acid solution with deionized water at 121 °C for 1 h, followed by filtration. The acid-insoluble lignin (AIL) content was measured gravimetrically after mineralization at 450 °C for 12 h. The acid-soluble lignin (ASL) content was determined in the liquid fraction on a Shimadzu UV-1800 spectrophotometer (Kyoto, Japan) at 240 nm using an absorptivity constant of 25 L/g·cm. The liquid fraction was neutralized with calcium carbonate from Merck (Darmstadt, Germany) until reaching neutral pH and filtered on a 0.2 µm cellulose acetate filter before HPLC analysis on a Thermo Ultimate 3000 HPLC system from Thermo Scientific (Sunnyvale, CA, USA). All HPLC standards (i.e., acetic acid, arabinose, glucose, and xylose) were purchased from Sigma-Aldrich (Saint-Louis, MO, USA). A Rezex RHM-Monosaccharide H+ 300 × 7.8 mm column connected to a Rezex RHM-Monosaccharide H+ 50 × 7.8 mm guard column, both from Phenomenex (Torrance, CA, USA), were used with 5 mmol/L of H_2_SO_4_ as an eluent at a rate of 0.6 mL/min. The injection volume was 50 µL, while the column was maintained at 65 °C and the RI detector at 50 °C.

#### 3.3.2. Chemical Composition of the Supernatant

Total supernatant concentrations were determined by DM content measurement, as previously described. The ash content was determined as for the solid samples. The supernatant was filtered on a 0.2 µm cellulose acetate filter before direct HPLC analysis to determine the acetic acid and monomeric sugar concentrations (i.e., arabinose, glucose, and xylose). One-step hydrolysis was also applied to the supernatants in triplicate to determine the total sugar concentrations (monomers and oligomers) and lignin contents. For this purpose, 10 mL of supernatant were placed in a hydrolysis tube with 362 µL of 72% sulfuric acid to achieve a final 4% sulfuric acid concentration. The solutions were autoclaved at 121 °C for 1 h, and then filtered and neutralized with calcium carbonate until reaching neutral pH. Lignin, hemicellulose, and cellulose contents were then determined according to the procedure described for solid samples in a previous section.

Total polyphenolic compounds were determined using the Folin–Ciocalteu colorimetric method, adapted from Singleton and Rossi [52], on all supernatants filtered using a 96-well microplate. Phenolic compounds were reacted with Folin reagent, i.e., consisting of a mixture of phosphotungstic acid and phosphomolybdic acid, in a reaction in which phenols were oxidized by tungsten and molybdenum oxide reduction, yielding a blue color. Polyphenol levels were measured by visible spectroscopy using a calibration curve. Gallic acid, used as reference standard; Folin–Ciocalteu reagent; and Na_2_CO_3_ were purchased from Sigma-Aldrich (Saint-Louis, MO, USA). Each well was successively filled with 20 µL of supernatant, 10 µL of Folin–Ciocalteu reagent, and 170 µL of a solution containing 2.36% Na_2_CO_3_. Each sample was analyzed eight times. After the addition of Folin–Ciocalteau reagent, the microplate was placed in a Spectrostar-Nano plate reader from BMG-Labtech (Champigny-sur-Marne, France) and shaken for 10 s. After the addition of Na_2_CO_3_, the plate was again placed in the reader and shaken for 10 s prior to the analysis. Absorbance was measured at a 700 nm wavelength at 45 °C after a 45 min reaction time. The final concentration was expressed in grams of gallic acid equivalent per liter (g GAE/L).

### 3.4. Physical Characterization of Extrudates

The particle size distributions of the raw SCB and extrudates were determined in triplicate using a Retsch AS 200 vibratory sieve shaker (Haan, Germany). Samples weighing 40 g were placed in the shaker for 10 min at 1 mm amplitude using a series of six sieves with 6.3 mm, 4.0 mm, 2.0 mm, 1.0 mm, 0.8 mm, and 0.2 mm mesh openings and a bottom plate. The amount of extrudate retained on each sieve was weighed and expressed as a weight percentage in proportion to the total weight. The fine particle fraction was defined as the fraction retained <0.2 mm and collected in the bottom plate.

SEM raw SCB and extrudate images were obtained using a FEI Quanta 450 scanning electron microscope (Hillsboro, OR, USA), with 130 Pa water vapor partial pressure in the chamber at high voltage (12.5 kW) without saturation of the samples.

The bulk and tapped raw SCB and extrudate densities were determined using a Granuloshop Densitap ETD-20 volumenometer (Chatou, France) fitted with a 250 mL graduated cylinder. The sample was weighed in the graduated cylinder and the volume was recorded prior to compaction to determine the bulk density. The cylinder was then tapped 500 times on the volumenometer at 3 mm height and 250 taps/min. The volume was measured at the nearest graduation and the operation was then repeated until a constant volume was obtained to determine the tapped density. All measurements were performed in triplicate.

### 3.5. Preparation of Binderless Materials by Thermocompression

All SCB materials were obtained by thermocompression of 20 g of raw SCB or extrudate using a steel mold. A 50 t capacity heated hydraulic press from Pinette Emidecau Industries (Chalon-sur-Saône, France) was used to produce flat square materials measuring 70 mm × 70 mm. A uniaxial pressure of 102 MPa was applied to the sample, and the mold was then heated to 200 °C. The temperature was maintained for 10 min, and finally the mold was cooled down for 10 min while maintaining the pressure before opening. For each tested extrusion condition, two materials were produced and photographed. The materials were cut into eight 45 mm-long × 10 mm-wide specimens, and then stored in an environmental chamber at 25 °C and 50% RH for 2 weeks until constant weight before assessing the properties.

### 3.6. Material Characterization

The density of the materials obtained after thermocompression was determined in triplicate using the remaining pieces of material obtained after cutting the bending test specimens. The densities of these samples were assessed using a method based on Archimedes’ principle, with cyclohexane as the immersion liquid. Using a Sartorius hydrostatic balance (Göttingen, Germany) capable of weighing in both air and liquid media, we were able to determine the density of our samples with the following formula:(1)$$Density=\frac{w_{air}*(d_{cyclohexane}-d_{air})}{\left(w_{air}-w_{cyclohexane}\right)*Corr}+d_{air}$$
where $w_{air}$ and $w_{cyclohexane}$ are the sample weights measured in air and cyclohexane, $d_{air}$ and $d_{cyclohexane}$ are the densities of air and cyclohexane at room temperature, and Corr is the thrust correction factor due to the submerged wire.

The bending properties of the test specimens were assessed according to the ISO 16978:2003 standard [53] using a Tinius Olsen universal testing machine (Horsham, PA, USA) fitted with a 500 N load cell and the three-point bending test. The thicknesses and widths were measured at the specimen’s center with a Tacklife electronic digital sliding caliper (Levittown, NY, USA). The testing speed was set at 1 mm/min, with a 40 mm separation grip. Bending properties were characterized by testing the sixteen specimens cut from the materials. The properties evaluated were the flexural modulus and flexural strength at breaking point, determined with the following formulas:(2)$$\mathrm{Flexural}\mathrm{modulus}=\frac{L^{3}}{4*w*t^{3}}*\frac{F_{2}-F_{1}}{d_{2}-d_{1}}$$
(3)$$\mathrm{Flexural}\mathrm{strength}=F_{breaking}*\frac{3*L}{2*w*t^{2}}$$
where L is the length between the two supports (40 mm); w is the sample width; t is the sample thickness; $F_{1}$ and $F_{2}$ are the forces measured for $d_{1}$ and $d_{2}$ deformations at 10% and 40% of $F_{breaking}$, respectively; and $F_{breaking}$ is the force measured at the breaking point.

Water resistance was tested by immersing 45 mm-long × 10 mm-wide specimens in water at 25 °C to determine the water absorption (WA) and thickness swelling (TS) in triplicate according to the ISO 16983:2003 standard [54]. Before soaking, samples were oven-dried at 105 °C until constant weight to ensure a uniform initial condition. The initial thickness and weight were then measured. The samples were subsequently submerged in distilled water at 25 °C for 24 h. The sample weights and thicknesses were measured hourly for the first 8 h and again after 24 h. The thickness of each sample was measured at three points, i.e., at the center and both ends. WA and TS were calculated for each sample at 24 h using the following formulas:(4)$$WA=\frac{w_{24h}-w_{0}}{w_{0}}*100$$
(5)$$TS=\frac{t_{24h}-t_{0}}{t_{0}}*100$$
where $w_{0}$ and $w_{24h}$ are the sample weights initially and after 24 h of water immersion, and $t_{0}$ and $t_{24h}$ are the sample thicknesses initially and after 24 h of water immersion.

### 3.7. Statistical Analysis

WA, TS, and density determinations were conducted in triplicate. For the mechanical properties, 16 samples were tested from each extrusion condition. Means were statistically compared by one-way analysis of variance (ANOVA) with α = 0.05 and Student’s *t*-test. In tables and figures presenting the results, values with no significant difference are presented with the same letter (a–d).

## 4. Conclusions

Continuous aqueous pretreatment of SCB was achieved by twin-screw extrusion. In the filtration configuration, increasing the L/S ratio to 2.05 significantly increased the extraction yield. Using a low L/S ratio (<1.25) resulted in stronger mechanical action, leading to increased extrudate density, fiber aggregation, and partial SCB degradation, resulting in a material with lower flexural strength. With an L/S ratio > 1.25, the water present facilitated molecule solubilization, and the materials obtained had higher mechanical properties. An L/S ratio of 2.05 was the best tradeoff between the extraction yield (11.5 g/kg DMi), water consumption, and material properties (1485 kg/m^3^ density, 6.2 GPa flexural modulus, 51.2 MPa flexural strength, and WA and TS values of 37% and 44%, respectively). Under these conditions, the pretreatment substantially improved the flexural strength by 54% and the flexural modulus by 27% and reduced the WA by 59% and TS by 56% due to fiber deconstruction. Finally, the extrudate obtained without liquid–solid separation yielded materials with properties equivalent to those of the extrudates obtained with liquid–solid separation for L/S ratios of at least 1.25, indicating that the partial removal of hemicelluloses and lignin was limited and did not markedly change the final material properties.

## Figures and Tables

**Figure 1 molecules-29-03020-f001:**
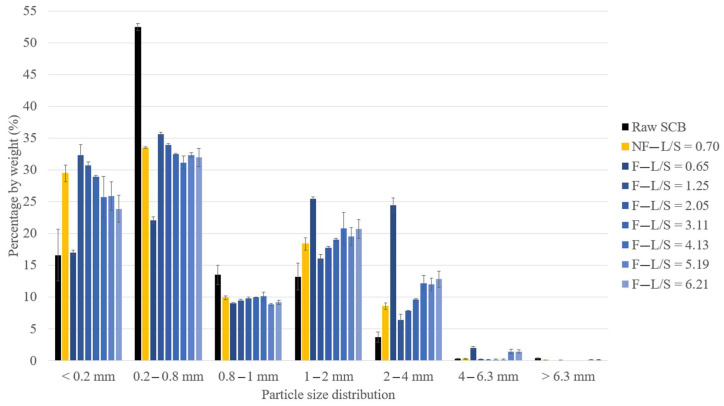
Particle size distribution of raw SCB and extrudates obtained at different L/S ratios using a Clextral Evolum HT 53 twin-screw extruder with and without a filtration module (error bars represent standard deviations).

**Figure 2 molecules-29-03020-f002:**
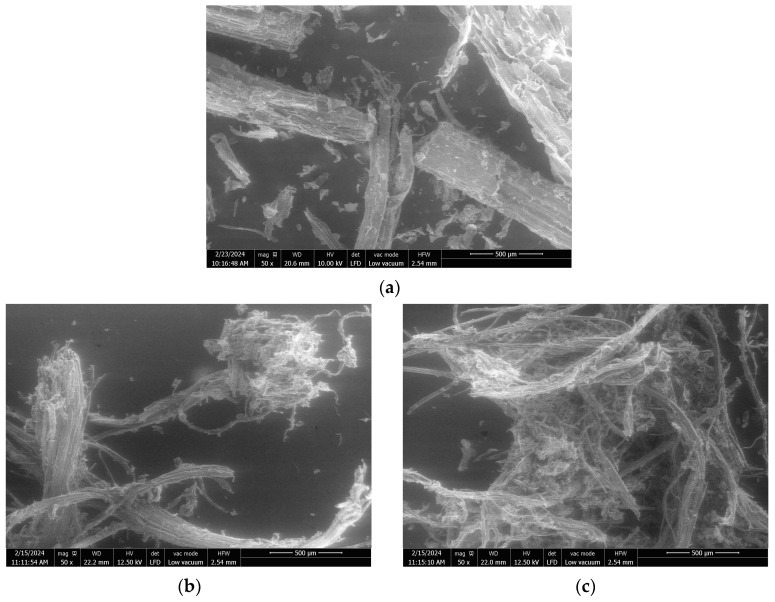
SEM images at ×50 magnification of raw SCB (**a**) and extrudates obtained using a Clextral Evolum HT 53 twin-screw extruder without filtration at an L/S ratio of 0.65 (**b**,**c**), and with filtration at an L/S ratio of 0.65 (**d**,**e**), 2.05 (**f**,**g**), and 6.21 (**h**,**i**).

**Figure 3 molecules-29-03020-f003:**
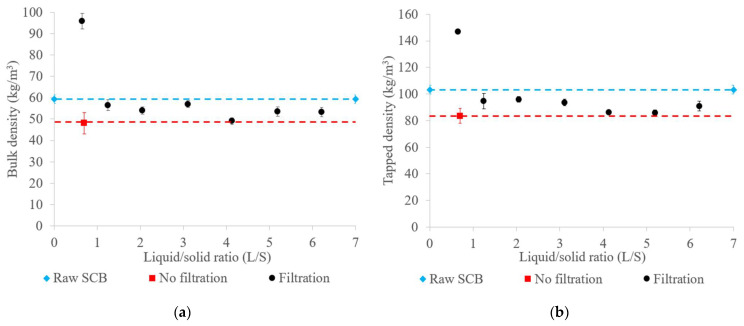
Bulk (**a**) and tapped densities (**b**) of raw SCB and extrudates obtained using a Clextral Evolum HT 53 twin-screw extruder with and without filtration at different L/S ratios. Error bars represent the standard deviation.

**Figure 4 molecules-29-03020-f004:**
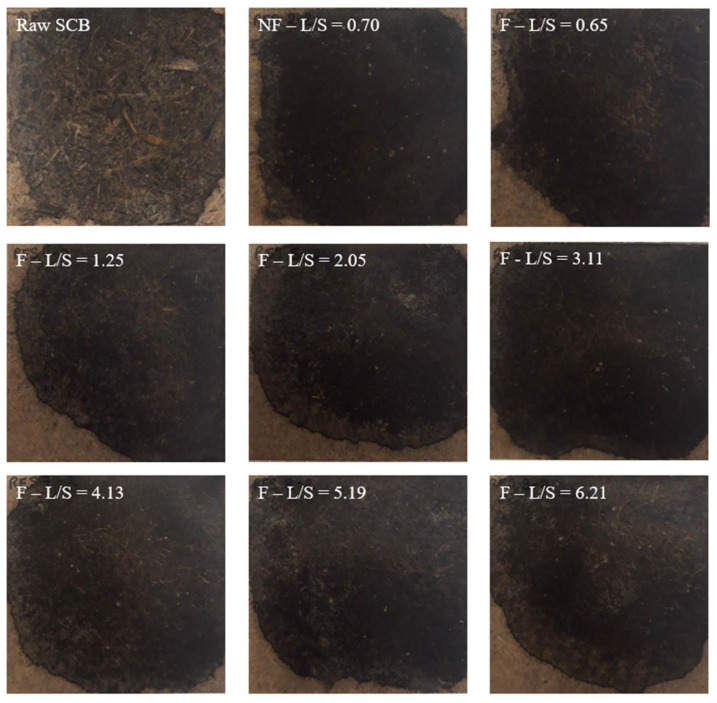
Pictures of the materials produced with raw SCB and extrudates obtained at different L/S ratios using a Clextral Evolum HT 53 twin-screw extruder equipped with and without a filtration module.

**Figure 5 molecules-29-03020-f005:**
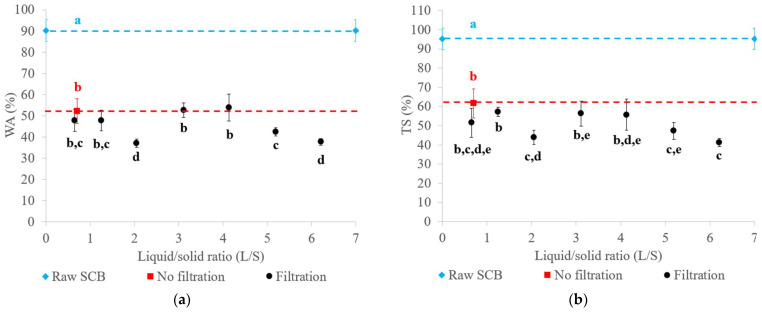
WA (**a**) and TS (**b**) of materials generated from raw SCB and extrudates obtained using a Clextral Evolum HT 53 twin-screw extruder equipped with and without filtration with different L/S ratios. Letters a–e refer to Student’s *t*-test results, and error bars represent the standard deviation.

**Figure 6 molecules-29-03020-f006:**
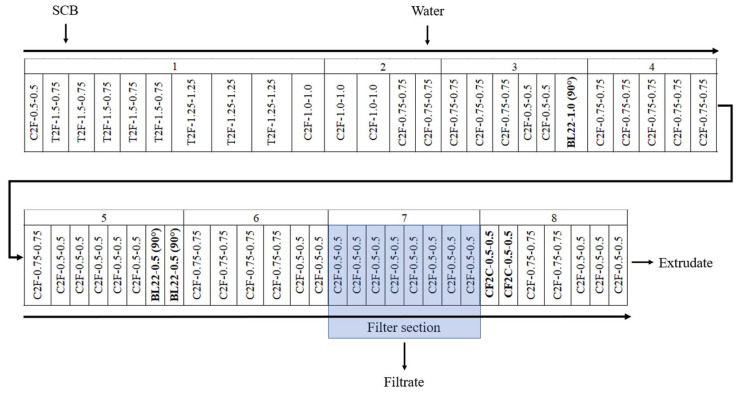
Screw profile configuration for the SCB aqueous pretreatment using a Clextral Evolum HT 53 twin-screw extruder. C2F: conjugated double-flight screws; T2F: trapezoidal double-flight screws; BL22: 2-lobe kneading blocks; CF2C: conjugated cut-flight, double-flight screws. The two numbers following the type of screw element indicate the pitch and length, respectively, of the C2F, T2F, and CF2C screws. The two numbers following the BB mixing blocks represent the length and staggering angle, respectively.

**Table 1 molecules-29-03020-t001:** Operating conditions used to fractionate SCB using a Clextral Evolum HT 53 twin-screw extruder, with and without a filtration module for extrudate and filtrate production, and the fractionation results.

Trials	1	2	3	4	5	6	7	8
Configuration	NF	F	F	F	F	F	F	F
**Input**								
Q_SCB_ (kg/h)	28.2	27.4	27.4	27.4	27.4	27.4	27.4	27.4
MC_SCB_ (%)	29.0	26.9	26.9	26.9	26.9	26.9	26.9	26.9
Q_SCB_ (kg DM/h)	20.0	20.0	20.0	20.0	20.0	20.0	20.0	20.0
Q_water pump_ (kg/h)	6.1	5.6	17.7	33.7	54.8	75.4	96.5	117.0
Q_L_ (kg/h)	14.3	13.0	25.1	41.1	62.2	82.8	103.9	124.4
Q_L_/Q_S_ (L/S ratio)	0.70	0.65	1.25	2.05	3.11	4.13	5.19	6.21
**Extrudate**								
Q_EXT_ (kg/h)	34.2 ± 0.9	27.4 ± 0.5	33.4 ± 0.5	29.6 ± 0.5	28.3 ± 0.3	27.4 ± 0.5	26.3 ± 0.1	24.8 ± 0.5
MC_EXT_ (%)	39.7 ± 0.6	24.3 ± 0.5	38.9 ± 0.3	34.4 ± 0.5	34.0 ± 1.7	33.0 ± 1.8	32.1 ± 0.1	31.7 ± 0.3
Q_EXT_ (kg DM/h)	20.6	20.7	20.4	19.4	18.7	18.3	17.8	16.9
Q_EXT water_ (kg/h)	13.6	6.7	13.0	10.2	9.6	9.0	8.4	7.8
**Filtrate**								
Q_FIL_ (kg/h)	n/a	0.4 ± 0.1	5.3 ± 0.4	24.7 ± 0.4	45.1 ± 0.3	65.8 ± 0.2	87.4 ± 0.8	96.2 ± 1.0
MC_FIL_ (%)	n/a	99.8 ± 0.1	93.4 ± 0.2	95.6 ± 0.4	96.6 ± 0.3	98.1 ± 0.1	98.6 ± 0.1	98.6 ± 0.9
Q_FIL_ (kg DM/h)	n/a	<0.1	0.3	1.1	1.6	1.2	1.2	1.4
Q_FIL water_ (kg/h)	n/a	0.4	4.9	23.6	43.5	64.5	86.2	94.8
**Global output**								
Q_EXT_ + Q_FIL_(kg DM/h)	20.6	20.7	20.7	20.5	20.2	19.6	19.1	18.3
Q_EXT water_ + Q_FIL water_(kg/h of water)	13.6	7.0	17.9	33.8	53.1	73.6	94.6	102.7
E_w_ (kg/h)	0.6	6.0	7.2	7.3	9.0	9.2	9.3	21.7
E_w_ (%)	4.2	46.0	28.6	17.8	14.5	11.1	9.0	17.4
I (A)	87 ± 4	94 ± 9	80 ± 7	80 ± 7	81 ± 6	81 ± 7	81 ± 5	79 ± 12
SME (W h/kg DM)	418 ± 19	436 ± 40	370 ± 30	368 ± 26	378 ± 30	376 ± 30	376 ± 21	359 ± 56
T_CF2C_ (°C)	115 ± 1	97 ± 1	98 ± 1	97 ± 1	92 ± 1	88 ± 1	86 ± 1	83 ± 2
T_Module8_ (°C)	110 ± 1	110 ± 1	110 ± 1	109 ± 1	108 ± 1	107 ± 1	106 ± 1	104 ± 1

Where NF is the configuration without filtration; F is the configuration with the filtration module; Q_SCB_ is the inlet SCB flow rate; MC_SCB_ is the SCB moisture content; Q_water pump_ is the inlet water flow rate from the pump; Q_L_ is the total inlet water flow rate (pump + SCB moisture); L/S is the liquid-to-solid ratio, defined as the ratio of the inlet water flow rate (Q_L_) to the inlet dry SCB flow rate (Q_SCB_); Q_EXT_ is the outlet extrudate flow rate; MC_EXT_ is the extrudate moisture content; Q_FIL_ is the outlet filtrate flow rate; MC_FIL_ is the filtrate moisture content; E_w_ is the estimated outlet evaporated water flow rate (kg/h and %, in proportion of the total inlet water flow rate); I is the current feeding the motor; SME is the specific mechanical energy relative to the inlet SCB dry matter flow rate; T_CF2C_ is the temperature of the mixture in the shearing area, i.e., in the vicinity of the reversed double-thread screws (CF2C); and T_Module8_ is the measured temperature of the last extruder module. N/a, not applicable.

**Table 2 molecules-29-03020-t002:** Distribution of dry matter in SCB filtrate between the soluble part (supernatant) and the solid particles (pellet) obtained during SCB fractionation using a Clextral Evolum HT 53 twin-screw extruder with a filtration module.

Trials	2	3	4	5	6	7	8
L/S ratio	0.65	1.25	2.05	3.11	4.13	5.19	6.21
Dry matter recovered in the filtrate (g/kg DMi)	<1	17 ± 4	55 ± 8	77 ± 6	62 ± 1	62 ± 3	69 ± 5
Dry matter recovered in the supernatant (%)	n/a	22	22	16	21	25	20
Dry matter recovered in the supernatant (g/kg DMi)	n/a	3.8	12.1	12.5	13.0	15.5	15.8
Dry matter recovered in the filtrate pellet (%)	n/a	78	78	84	79	75	80

**Table 3 molecules-29-03020-t003:** Concentrations and extraction yield in SCB supernatants as a function of the L/S ratio during SCB fractionation using a Clextral Evolum HT 53 twin-screw extruder with a filtration module.

Trials	3	4	5	6	7	8
L/S ratio	1.25	2.05	3.11	4.13	5.19	6.21
Concentration (g/L)	17.0 ± 0.1	11.3 ± 0.1	6.2 ± 0.1	4.1 ± 0.2	3.7 ± 0.1	3.3 ± 0.1
Total extraction yield (g/kg DMi)	3.7 ± 0.7	11.5 ± 0.9	11.8 ± 1.0	12.3 ± 0.9	14.5 ± 1.5	13.5 ± 0.8
Ash (g/kg DMi)	0.4 ± 0.1	1.3 ± 0.1	1.5 ± 0.2	1.8 ± 0.1	2.0 ± 0.5	2.2 ± 0.2
Glucose (g/kg DMi)	0.7 ± 0.1	2.0 ± 0.2	2.0 ± 0.1	2.1 ± 0.2	2.4 ± 0.1	1.9 ± 0.1
Xylose (g/kg DMi)	0.9 ± 0.1	2.7 ± 0.2	2.7 ± 0.3	2.8 ± 0.2	3.1 ± 0.2	2.7 ± 0.1
Arabinose (g/kg DMi)	0.3 ± 0.1	0.9 ± 0.1	0.9 ± 0.1	0.9 ± 0.1	1.0 ± 0.1	0.9 ± 0.1
Acetic acid (g/kg DMi)	0.2 ± 0.1	0.4 ± 0.1	0.4 ± 0.1	0.4 ± 0.1	0.5 ± 0.1	0.5 ± 0.1
Lignin (g/kg DMi)	1.0 ± 0.1	3.6 ± 0.1	3.4 ± 0.1	3.0 ± 0.1	4.4 ± 0.4	4.3 ± 0.1
Total polyphenols (g/kg DMi)	0.2 ± 0.1	0.6 ± 0.1	0.9 ± 0.1	1.3 ± 0.1	1.1 ± 0.1	1.0 ± 0.1

**Table 4 molecules-29-03020-t004:** Chemical compositions of raw SCB and SCB filtrate pellets as a function of the L/S ratio implemented for twin-screw extrusion using a Clextral Evolum HT 53 twin-screw extruder with a filtration module (% DM).

Trials	Raw SCB	3	4	5	6	7	8
L/S ratio	-	1.25	2.05	3.11	4.13	5.19	6.21
Dry matter (%)	91.5 ± 0.1	92.1 ± 0.2	93.1 ± 0.1	93.3 ± 0.1	93.0 ± 0.1	93.6 ± 1.9	94.2 ± 1.2
Ash (%)	5.1 ± 0.2	22.3 ± 0.3	24.6 ± 0.3	25.6 ± 0.2	27.6 ± 0.2	28.5 ± 0.5	24.0 ± 1.1
AIL (%)	19.3 ± 0.2	21.0 ± 1.3	17.3 ± 0.5	17.9 ± 0.5	24.1 ± 1.7	19.3 ± 0.7	18.6 ± 0.2
ASL (%)	4.8 ± 0.1	4.1 ± 0.3	3.8 ± 0.1	4.0 ± 0.1	3.8 ± 0.3	3.9 ± 0.1	3.7 ± 0.1
Total lignin (%)	24.1 ± 0.3	25.1 ± 1.2	21.2 ± 0.3	21.9 ± 0.5	28.0 ± 1.1	23.3 ± 0.5	22.3 ± 0.6
Cellulose (%)	40.6 ± 0.3	20.7 ± 1.1	21.3 ± 1.0	20.8 ± 0.4	16.2 ± 1.6	15.4 ± 1.2	22.6 ± 1.3
Xylose (%)	20.9 ± 0.1	12.1 ± 0.5	12.5 ± 0.6	12.1 ± 0.2	10.6 ± 0.2	8.8 ± 0.6	13.3 ± 0.8
Arabinose (%)	2.1 ± 0.1	2.0 ± 0.1	2.1 ± 0.1	2.2 ± 0.1	1.9 ± 0.1	1.7 ± 0.1	2.4 ± 0.1
Hemicelluloses (%)	23.0 ± 0.1	14.1 ± 0.5	14.6 ± 0.6	14.3 ± 0.2	12.5 ± 0.2	10.5 ± 0.6	15.7 ± 0.8
Acetyl (%)	3.9 ± 0.1	2.0 ± 0.1	2.0 ± 0.1	1.9 ± 0.1	1.7 ± 0.1	1.4 ± 0.1	2.1 ± 0.1
Extractables (H_2_O) (%)	2.1 ± 0.1	9.6 ± 0.7	11.3 ± 0.5	8.8 ± 0.4	7.7 ± 0.2	11.5 ± 0.1	8.6 ± 0.2
Extractables (EtOH) (%)	1.6 ± 0.7	3.5 ± 0.2	2.5 ± 0.1	2.5 ± 0.1	3.1 ± 0.1	3.3 ± 0.1	2.7 ± 0.1
Total (%)	100	101	101	100	100	101	100

**Table 5 molecules-29-03020-t005:** Chemical composition of raw SCB and extrudates obtained after aqueous pretreatment with a Clextral Evolum HT 53 twin-screw extruder (% DM).

Trials		1	2	3	4	5	6	7	8
L/S ratio	Raw SCB	0.70	0.65	1.25	2.05	3.11	4.13	5.19	6.21
Configuration	-	NF	F	F	F	F	F	F	F
DM (%)	91.5 ± 0.1	92.9 ± 0.2	93.1 ± 0.1	92.3 ± 0.1	92.1 ± 0.5	93.6 ± 0.7	93.0 ± 0.7	93.7 ± 0.6	92.0 ± 0.2
Ash (%)	5.1 ± 0.2	8.0 ± 0.1	9.1 ± 0.3	6.5 ± 0.9	7.3 ± 1.2	7.5 ± 0.1	6.1 ± 0.1	5.1 ± 0.3	5.8 ± 0.6
AIL (%)	19.3 ± 0.2	20.2 ± 0.9	20.2 ± 0.5	19.3 ± 0.5	19.7 ± 0.3	19.4 ± 0.2	18.8 ± 0.4	19.3 ± 0.4	19.4 ± 0.4
ASL (%)	4.8 ± 0.1	3.8 ± 0.1	3.6 ± 0.1	4.1 ± 0.1	4.3 ± 0.1	4.1 ± 0.1	4.2 ± 0.3	4.2 ± 0.1	4.1 ± 0.1
Total lignin (%)	24.1 ± 0.3	24.0 ± 0.9	23.8 ± 0.5	23.4 ± 0.6	24.0 ± 0.2	23.5 ± 0.2	23.0 ± 0.2	23.5 ± 0.3	23.5 ± 0.5
Cellulose (%)	40.6 ± 0.3	38.6 ± 0.3	37.1 ± 0.3	39.8 ± 1.0	40.0 ± 0.5	39.1 ± 2.2	39.7 ± 0.8	40.9 ± 0.5	40.6 ± 0.3
Xylose (%)	20.9 ± 0.1	19.7 ± 0.1	19.4 ± 0.1	20.6 ± 0.5	20.6 ± 0.3	20.2 ± 0.9	20.7 ± 0.2	21.3 ± 0.1	21.2 ± 0.2
Arabinose (%)	2.1 ± 0.1	1.9 ± 0.1	2.1 ± 0.1	2.1 ± 0.1	2.0 ± 0.1	2.0 ± 0.1	2.1 ± 0.1	2.1 ± 0.1	2.2 ± 0.1
Hemicelluloses (%)	23.0 ± 0.1	21.6 ± 0.1	21.5 ± 0.1	22.7 ± 0.2	22.6 ± 0.2	22.2 ± 0.8	22.8 ± 0.2	23.4 ± 0.1	23.4 ± 0.1
Acetyl (%)	3.9 ± 0.1	3.9 ± 0.4	3.6 ± 0.1	3.8 ± 0.1	3.9 ± 0.1	3.8 ± 0.1	3.8 ± 0.1	3.9 ± 0.1	3.9 ± 0.1
Extractables (H_2_O) (%)	2.1 ± 0.1	4.0 ± 0.2	3.7 ± 0.2	3.5 ± 0.2	2.5 ± 0.3	3.2 ± 0.1	2.7 ± 0.3	3.2 ± 0.4	1.8 ± 0.1
Extractables (EtOH) (%)	1.6 ± 0.7	0.7 ± 0.1	1.1 ± 0.1	1.1 ± 0.1	1.1 ± 0.1	1.0 ± 0.1	1.6 ± 0.1	1.3 ± 0.1	1.4 ± 0.2
Total (%)	100	101	100	101	101	100	100	101	100

**Table 6 molecules-29-03020-t006:** Density and mechanical properties of materials produced with SCB extrudates. Letters a–e refer to Student’s *t*-test results.

Trials	Configuration	L/S Ratio	Density (kg/m^3^)	Flexural Strength (MPa)	Flexural Modulus (GPa)
	Raw SCB		1432 ± 16 ^a^	33.2 ± 5.0 ^a^	4.9 ± 0.4 ^a^
1	No filtration	0.70	1498 ± 15 ^b^	53.0 ± 3.7 ^c,d^	6.6 ± 0.7 ^b,c,d,e^
2	Filtration	0.65	1503 ± 10 ^b^	38.2 ± 4.2 ^b^	6.1 ± 0.7 ^b^
3		1.25	1496 ± 20 ^b,c^	53.7 ± 4.9 ^c,d^	6.5 ± 0.5 ^b,c^
4		2.05	1485 ± 9 ^c,d^	51.2 ± 5.3 ^c^	6.2 ± 0.8 ^b,d^
5		3.11	1481 ± 16 ^c,d,e^	54.5 ± 3.0 ^d^	6.3 ± 0.5 ^b,d^
6		4.13	1482 ± 9 ^d,e^	54.2 ± 4.8 ^c,d^	6.6 ± 0.4 ^c,d^
7		5.19	1480 ± 9 ^d,e^	55.7 ± 3.8 ^d^	6.8 ± 0.5 ^c,e^
8		6.21	1475 ± 10 ^e^	56.0 ± 3.7 ^d^	7.1 ± 0.6 ^e^

## Data Availability

The raw data supporting the conclusions of this article will be made available by the authors upon request.

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
