# Peer review of "Aqueous Pretreatment of Lignocellulosic Biomass for Binderless Material Production: Influence of Twin-Screw Extrusion Configuration and Liquid-to-Solid Ratio"

_molecules, 2024, doi:10.3390/molecules29133020_

Round 1

Reviewer 1 Report

Comments and Suggestions for Authors

The authors have presented their research manuscript entitled, “Aqueous Pretreatment of Lignocellulosic Biomass for Binder-less Material Production: Influence of Twin-Screw Extrusion Configuration and Liquid-to-Solid Ratio.” Overall, the manuscript is well written with good presentation of data and hypothesis. The manuscript needs addressal of minor points before being considered for publication in ‘Molecules’ journal.

1.     The novelty and significance of the review work must be presented in the abstract and introduction section. Also add a take home message (concluding statement) of the manuscript at the end of the abstract.

2.     Section 2.2.2., 2.4.1: Please add latest 1-2 discussions in these sections, respectively, comparing your results with previous literature.

3.     Line 245: why there is missing citation? Error? Kindly rectify this.

4.     Line 450: Table 1 is written twice.

5.     Conclusion section should be limited to 150 words only, highlighting major outcomes.

Author Response

Comment 1: The novelty and significance of the review work must be presented in the abstract and introduction section. Also add a take home message (concluding statement) of the manuscript at the end of the abstract

Response 1: Thank you for this comment. We added some elements to present the novelty and significance of the work in the abstract (lines 12 to 14) and introduction (lines 104 to 107). We also added a concluding statement at the end of the abstract (lines 27 to 29).

Comment 2: Section 2.2.2., 2.4.1: Please add latest 1-2 discussions in these sections, respectively, comparing your results with previous literature.

Response 2: Thank you for this comment. The first discussion has been added by comparing the extraction results with the previous literature for point 2.2.2. from lines 205 to 214. The second one has been added by comparing the mechanical properties with the previous literature for point 2.4.1. from lines 375 to 386 in the revised version and the water resistance for point 2.4.2. from lines 410 to 419.

Comment 3: Line 245: why there is missing citation? Error? Kindly rectify this.

Response 3: Thank you for pointing this out. We have rectified this error and the missing reference corresponded to Figure 1 in line 259.

Comment 4: Line 450: Table 1 is written twice.

Response 4: Thank you for pointing this out. We have deleted the second reference of Table 1 in line 492.

Comment 5: Conclusion section should be limited to 150 words only, highlighting major outcomes.

Response 5: Thank you for your comment. We have reduced the conclusion section under 200 words, from lines 636 to 651.

Reviewer 2 Report

Comments and Suggestions for Authors

The experimental article “Aqueous Pretreatment of Lignocellulosic Biomass for Binderless Material Production: Influence of Twin-Screw Extrusion Configuration and Liquid-to-Solid Ratio” is dedicated to investigating the continuous aqueous pretreatment of sugarcane bagasse (SCB) through twin-screw extrusion. The authors tested two configurations with and without a filter module, and investigated the effect of the liquid-solid ratio on the properties of the resulting material, as well as the properties of materials obtained by thermopressing the extrudate. The authors know the terminology and are well versed in this topic, so the article is well written and can be published in the journal Molecules. However, the reviewer had some minor recommendations that needed to be addressed.

1. In the introduction there is only one reference from 2024. It is recommended to enhance the relevance of the work by including more recent publications (2022-2024) on this topic in the introduction.

2. Question about the practical application of the obtained materials. The authors conclude that the obtained materials can be used to make furniture. However, the materials do not meet the criteria for use in wet environments or outdoors. Can these materials be used or not? I would like to see more specific recommendations on this topic, incl. in paragraph 4. Conclusion.

Author Response

Comment 1: In the introduction there is only one reference from 2024. It is recommended to enhance the relevance of the work by including more recent publications (2022-2024) on this topic in the introduction.

Response 1: Thank you for your advice. We modified the introduction to add more recent publications  in lines 43-44 and  line 89.

Comment 2: Question about the practical application of the obtained materials. The authors conclude that the obtained materials can be used to make furniture. However, the materials do not meet the criteria for use in wet environments or outdoors. Can these materials be used or not? I would like to see more specific recommendations on this topic, incl. in paragraph 4. Conclusion.

Response 2: Thank you for your comment. We added a paragraph in the section 2.4.3. to discuss this point and gave different recommendations on this topic in lines 443 to 450.